# Cerium Oxide Nanoparticles Regulate Oxidative Stress in HeLa Cells by Increasing the Aquaporin-Mediated Hydrogen Peroxide Permeability

**DOI:** 10.3390/ijms231810837

**Published:** 2022-09-16

**Authors:** Giorgia Pellavio, Patrizia Sommi, Umberto Anselmi-Tamburini, Maria Paola DeMichelis, Stefania Coniglio, Umberto Laforenza

**Affiliations:** 1Department of Molecular Medicine, Human Physiology Unit, University of Pavia, 27100 Pavia, Italy; 2Department of Chemistry, University of Pavia, 27100 Pavia, Italy

**Keywords:** CeO_2_, HeLa, nanoparticles, water channels, peroxiporin, HyPer7 probe

## Abstract

Some aquaporins (AQPs) allow the diffusion of hydrogen peroxide (H_2_O_2_), the most abundant ROS, through the cell membranes. Therefore, the possibility of regulating the AQP-mediated permeability to H_2_O_2_, and thus ROS scavenging, appears particularly important for controlling the redox state of cells in physiological and pathophysiological conditions. Several compounds have been screened and characterized for this purpose. This study aimed to analyze the effect of cerium oxide nanoparticles (CNPs) presenting antioxidant activity on AQP functioning. HeLa cells express AQP3, 6, 8, and 11, able to facilitate H_2_O_2_. AQP3, 6, and 8 are expressed in the plasma membrane and intracellularly, while AQP11 resides only in intracellular structures. CNPs but not cerium ions treatment significantly increased the water and H_2_O_2_ permeability by interacting with AQP3, 6, and especially with AQP8. CNPs increased considerably the AQP-mediated water diffusion in cells with oxidative stress. Functional experiments with silenced HeLa cells revealed that CNPs increased the H_2_O_2_ diffusion mainly by modulating the AQP8 permeability but also the AQP3 and AQP6, even if to a lesser extent. Current findings suggest that CNPs represent a promising pharmaceutical agent that might potentially be used in numerous pathologies involving oxidative stress as tumors and neurodegenerative diseases.

## 1. Introduction

Aquaporins (AQPs) are a superfamily of channel-forming proteins that facilitate the diffusion of water and various small solute such as glycerol and ammonia [1,2]. In addition to the classic role of AQPs in facilitating the transmembrane diffusion of water in response to an osmotic gradient, they are involved in other unexpected physiological processes: skin hydration, elasticity, and wound healing; triglycerides accumulation in adipocytes; liver gluconeogenesis and detoxification; neural signal transduction (despite the absence of AQPs in neurons); and cell migration [3,4,5]. As a result, AQPs are involved in a large number of noninfectious diseases including obesity and metabolic syndrome, cancer, skin diseases, neurological disorder and epilepsy, and immune cell dysfunction [6,7]. 

Seven AQP paralogs (AQP0, 1, 3, 5, 8, 9, and AQP11) have shown hydrogen peroxide (H_2_O_2_) permeability and, for this reason, are now called peroxiporins [8,9,10,11,12,13,14,15,16,17,18]. The control of peroxiporin-mediated permeability to H_2_O_2_ appears to be of great importance in regulating the redox state of cells. At low, physiologic concentrations, H_2_O_2_ acts as a signaling molecule but higher concentrations may cause oxidative stress, which may trigger programmed cell death. Recently, AQP-mediated H_2_O_2_ diffusion was found to be reduced by different stress conditions [8,19,20,21], so this could be defined as a negative feedback control between the H_2_O_2_ levels and the permeability of AQPs. This negative effect of supraphysiological levels of H_2_O_2_ on AQP-mediated permeability may worsen the cell oxidative stress from mild to harsh leading to growth arrest and cell death [8,15]. In this scenario, AQPs represent an important antioxidant system regulating the H_2_O_2_ transport and thus cell signaling and survival during stress. 

Aquaporins and peroxiporins are now widely studied as potential druggable targets for a large number of pathologic conditions such as cancer and degenerative diseases [7]. Many compounds were screened for their ability to modulate the pore gating of the AQPs and some of them resulted in being effective [7,22,23]. They could be chemically classified into three groups: small molecule inhibitors, heavy metal ion inhibitors, and antibodies [23]. Among the small molecules active in the modulation of AQPs, some food nutrients, phytochemical compounds, and derivatives also displayed antioxidant effects [20,22,24,25]. More recently, sigma1 receptor agonists were shown to restore the AQP-mediated water permeability in the heat-stressed HeLa cells [26]. Although several compounds have been identified to inhibit or normalize the permeability of AQPs, so far, nothing has been found capable of increasing the AQP efficiency. 

Cerium oxide nanoparticles or ceria (CNPs) have gained increasing interest thanks to their significant antioxidant activity. CNPs represent a promising pharmaceutical agent that might potentially be used in all pathologies associated with a high level of free radicals. Although their effects have been extensively documented [27,28,29,30,31,32,33], only a few studies have investigated the mechanism behind their action as free radical scavengers [34].

Most of the proposed mechanisms are based on data obtained in abiotic experiments and suggest that the antioxidant activity of ceria relies on the balance between the two oxidation states of cerium (Ce(III) and Ce(IV)) present on the surface of CNPs. Such a dynamic equilibrium would generate an oxidation–reduction cycle making the CNP a self-regenerating oxidant scavenger. Experiments performed in biological environments showed that the Ce(III)/Ce(IV) ratio was strongly dependent not only on the CNPs intrinsic characteristics but also on their intracellular localization [34]. It has also been evidenced that ultrasmall CNPs functionalized with polyacrylic acid (PAA) present a very strong affinity towards the cell membrane; resulting in a very rapid interaction even in conditions inhibiting active cellular incorporation mechanisms [35].

Herein, we investigated a possible relationship between CNPs and AQPs activity, considering whether NPs could somehow modify the activity of AQPs as controllers of the redox state of the cell. HeLa cells were used as a cell model as they were already used in our previous studies, and they are well characterized in terms of AQPs expression and function and have been utilized in other CNPs studies [8,20,26,34,35].

To this purpose, we investigated the CNPs effect on the AQP-mediated water and H_2_O_2_ permeability. In particular, we considered all the peroxiporins expressed by HeLa cells and AQP6 whose permeability to H_2_O_2_ was recently demonstrated [36]. We examined: (1) the effect of CNPs on water permeability of HeLa cells in the absence and in the presence of oxidative stress conditions; (2) the possible interaction between AQPs and CNPs by confocal fluorescence microscopy; (3) the effect of CNPs on the H_2_O_2_ permeability of Hela cells silenced for single specific AQPs. 

The results reported here provide evidence that supersmall CNPs (5 nm in size) exert an increase in the AQP-mediated permeability to H_2_O_2_ and, for the first time, the possibility of chemically increasing the AQP pore gating. This is a promising step toward the development of innovative treatment of diseases involving oxidative stress such as cancer and degenerative disorders. 

## 2. Results

### 2.1. Aquaporins mRNA and Protein Expression in HeLa Cells

The expression of AQP1–AQP11 was investigated by qRT-PCR in HeLa cells and the results demonstrated the presence of all transcripts except those of AQP2 and AQP10 (Figure 1). The PCR products were checked by agarose gel electrophoresis and the results showed single bands of the expected size (not shown). The transcript levels were expressed as ΔCt, so high ΔCt values indicate low transcript expression contents. As shown, AQP6, AQP3, and AQP9 transcripts were more abundant than those of AQP7, AQP5, AQP8, AQP11, and AQP4.

The AQP expression was then examined at the protein level by immunoblotting and immunofluorescence. Figure 2 shows the blots of AQPs whose protein expression was also confirmed by immunocytochemistry: AQP3, AQP6, AQP8, and AQP11. 

Major bands were observed approximately at 31 kDa. AQP3 and AQP6 showed an additional band at 62 kDa probably representing the dimer form, and (AQP3) a band probably representing the glycosylated form. The sizes of the bands confirmed our previous findings and were in agreement with data already reported in the literature. The expression of the housekeeping gene B2M was also shown (Figure 2). 

### 2.2. CNPs and Aquaporin-3, -6, -8, and -11 Localization in HeLa Cells

The cellular colocalization of AQP3, AQP6, AQP8, and AQP11 with CNPs was investigated in HeLa cells by double-label immunofluorescence. 

Results showed localization of AQP3, AQP6, and AQP8 in the plasma membrane and intracellular structures while AQP11 was almost exclusively intracellular (green labeling; Figure 3A,D). CNPs localized on the plasma membrane and inside the cells (Figure 3B,D). Negative controls were also performed in the absence of antibodies by incubating with nonimmune serum and did not show any staining (Appendix A).

To evaluate a possible colocalization of the CNPs with the single AQPs, we analyzed 3D images using JACoP from Fiji and quantified the Pearson’s correlation coefficient r, Van Steensel’s cross-correlation function (CCF), and Manders’ colocalization coefficients (M1 and M2) (Figure 4). 

Pearson’s correlation coefficients r for AQP3 and AQP8 were higher than 0.7, supporting a possible colocalization with CNPs [37]. For AQP6 and AQP11, the r coefficients were significantly lower and had values of about 0.5, which did not strongly support their colocalization with CNPs (Figure 4A). Van Steensel’s CCFs were calculated to clarify this point [38]. Results showed CCFmax ranging from 0.68 to 0.84 for AQP3, AQP8, and AQP6 suggesting a possible colocalization with CNPs, while the CCFmax for AQP11 of about 0.5 did not clarify its possible interaction with CNPs (Figure 4B). As for Manders’ overlap coefficients M1 and M2, they indicated, respectively, the percentage of the single AQP (green signal) coincident with a CNP signal (red) over its total intensity and vice versa [39]. The results showed that the Manders’ coefficient M1 did not differ significantly for AQP3, AQP6, and AQP8, indicating that about 40% of the AQP bound CNP. M1 was significantly lower for AQP11, the amount of AQP11 bound to CNPs being only 25% of total AQP11 (Figure 4C). Finally, Manders’ coefficient M2 suggested a higher percentage of binding of CNPs with AQP3 (about 60%; Figure 4D). The amount of binding (relative proportion) of CNPs with the AQPs was the following: AQP3 >> AQP8 = AQP11 > AQP6.

As a whole, AQP3, AQP6, and AQP8 seemed to interact with CNPs, while the effect of AQP11 was not clearly demonstrated. 

### 2.3. The Effect of CNPs on Water Permeability of HeLa Cells in Normal and Oxidative Stress Conditions

The possible antioxidant effect of CNPs by modulating the AQP-mediated H_2_O_2_ efflux was investigated by measuring the water permeability. The water permeability to AQPs was demonstrated to be indicative of that of H_2_O_2_ [8]. After 15 min or 2 h, the cells were exposed to a 150 mOsm/L osmotic gradient. The treatment of cells with CNPs caused a reduction of the osmotic water permeability of about 50% at 15 min and an increase of about 60% after 2 h of incubation (Figure 5A). 

Usually, the effect of substances that inhibit the AQP-mediated water permeability was restored by subsequent incubation with β-mercaptoethanol or other reducing agents because AQP inhibitors act by oxidizing cysteine residues. Surprisingly, the post-treatment with an excess reducing agent could not restore the osmotic permeability to normal levels. This may be explained by the possible interaction of the reducing agents with the CNPs (Figure 5). 

To clarify whether the observed effects were due to the nanoparticles or just to their possible release of cerium ions, experiments were conducted by incubating the cells with a solution of cerium(III) nitrate. The results shown in Figure 5B demonstrate that cerium ions after both 15 min and 2 h of incubation decreased the water permeability by 25–30% and the subsequent treatment with an excess of β-mercaptoethanol further decreased the osmotic permeability, reaching a value equal to 50% of the control.

The effect of CNPs on the AQP-mediated water permeability was successively evaluated in oxidative stress conditions by treating cells with H_2_O_2_ (exogenous stress) or by heat stress (endogenous stress). As previously observed by our group, both treatments can decrease significantly the water permeability [8,20,40] (Figure 6). In the case of incubation with CNPs, after 2 h the water permeability increased (as also shown in Figure 5A), but the increment was even more relevant, and statistically significant, in both oxidative stress conditions (Figure 6).

Collectively, the results demonstrated that, after 2 h incubation, CNPs but not Ce(NO_3_)_3_ increased the water permeability of HeLa cells, especially in oxidative stress conditions. However, at a short incubation time (15 min) the water permeability was reduced by the treatment with both cerium nitrate and CNPs.

### 2.4. Effect of CNPs in HeLa Cells with Reduced Expression of Single AQPs

To understand which AQP (or AQPs) was the target of the antioxidant effect of CNPs, HeLa cells were selectively silenced for AQP3, AQP6, AQP8, and AQP11. Silencing was performed with either a siRNA targeting or a scrambled (i.e., control) siRNA. The efficacy of the silencing was determined by western blotting and the KO-cells were used 24 (Appendix A) and 48 (Appendix A) hours after transfection. An immunoblot analysis showed that after siRNA transfection, the level of AQP3, AQP6, AQP8, and AQP11 proteins decreased by 50, 50, 79, and 58%, respectively (Appendix A).

HeLa cells with reduced expression of AQPs and mock-transfected cells were then used to measure the time-course transport of H_2_O_2_ in the presence and the absence of CNPs treatment. Cells with AQP3 reduced expression showed a reduced H_2_O_2_ transport of about 57% (compared to the mock-transfected cells), but when cells were treated with CNPs the H_2_O_2_ transport increased proportionally in mock and silenced cells (Figure 7).

AQP6-silenced HeLa cells surprisingly showed transport of H_2_O_2_ doubled compared to mocked cells, but the treatment with CNPs was able to further increase the amount transported (Figure 8). To understand this result, AQP6-silenced cells were analyzed by immunoblotting for AQP3, 8, and 11. The results showed that HeLa cells had a compensatory upregulation of AQP8 (protein levels were doubled) (Appendix A). Cells silenced for AQP8 showed a greatly reduced H_2_O_2_ permeability of about 75% and the treatment with CNPs failed to enhance the amount transported (Figure 9). The reduced expression of AQP11 did not modify the plasma membrane permeability to H_2_O_2_ nor did the effect of CNPs (Figure 10).

Gene silencing demonstrated the leading role of plasma membrane AQP8 in mediating the CNPs effect, while a lower effect of CNPs on AQP3 and AQP6 emerged. 

## 3. Discussion

Hydrogen peroxide is the most abundant ROS in living cells and its accumulation leads to oxidative stress [8,9,41,42,43]. H_2_O_2_ at low concentrations can act as a signaling molecule in physiologic processes [8,44]. When oxidative stress occurs, cells use two ways to eliminate H_2_O_2_: through intracellular antioxidant systems or by promoting its efflux through the plasma membrane by means of some AQPs called peroxiporins. The modulation of peroxiporin-mediated H_2_O_2_ permeability seems, therefore, to have great importance in the regulation of the cell signaling pathway and in the survival from oxidative stress [8,19,20,21]. Furthermore, it has been observed that oxidative stress, experimentally induced by conditions such as heat or incubation with H_2_O_2_, causes a reduction in the permeability to H_2_O_2_ mediated by AQPs [8,19,20,21]. Recently, a number of compounds have been identified as capable of modulating AQPs by restoring their permeability in case of oxidative stress or by preventing the reduction of the permeability [20,24,26]. These findings open a new direction for the development of novel therapeutic treatments able to modulate the cellular signaling pathway and cell survival during oxidative stress in normal and pathological conditions, such as cancer and degenerative diseases [45,46]. The possibility of regulating the “gating” of the pores of peroxiporins has considerably accelerated the scientific research in this direction. Different types of modulators have been studied, such as metals, small organic molecules (drugs), natural compounds, microRNAs (miRNAs), hormones, peptides, and antibodies [47], all able to decrease the permeability but not increase it.

The aim of this study was to investigate nanoparticles as a possible new typology of AQPs modulator. Specifically, the attention was focused on ultrasmall cerium nanoparticles functionalized with polyacrylic acid [34]. Cerium nanoparticles have shown clear antioxidant properties capable of counteracting oxidative damage induced by radiation or inflammatory processes [31,48]. It has been observed that some formulations of nanoceria are instead able to induce the apoptosis of malignant lung cancer cells and favor the proliferative activity of stem cells in culture [49,50]. 

The expression of transcript and protein AQPs in HeLa cells was investigated and the results showed that the cells expressed the peroxiporins AQP3, AQP8, and AQP11, and the orthodox aquaporin AQP6. The latter is an AQP with peculiar functional properties: a low permeability at pH 7.0–7.4, a high permeability to water and anions at acid pH, and in the presence of Hg^+2^ in the external medium [51]. Recently, this AQP has also showed H_2_O_2_ permeability [36].

The possible interaction of the CNPs with the single AQPs was evaluated by confocal immunofluorescence and the analysis of 3D images (Figure 4). The values of Pearson’s, and Van Steensel’s coefficients revealed that AQP3, AQP8, and AQP6 interacted with CNPs, while the possible interaction of CNPs with AQP11 seemed uncertain or only partial. Manders’ coefficients indicated that CNPs bound AQPs in a variable percentage AQP3 >> AQP8, AQP11 > AQP6. It is interesting to observe how CNPs act in different cell domains, all involved in cellular redox regulation. AQP3, AQP6, and AQP8 are localized in the plasma membrane and intracellularly, and AQP11 is at the endoplasmic reticulum level [10,19,21,26]. It could therefore be speculated that CNPs act as antioxidants by promoting the elimination of H_2_O_2_ from the plasma membrane and by counteracting the endoplasmic reticulum stress. 

The CNPs free radical scavenger activity has been proposed to be comparable to that of physiological antioxidants, such as superoxide dismutase (SOD) and catalase [52,53]. However, differently from enzymes, CNPs possess a regenerative behavior. The spontaneous and reversible switching between Ce(IV) to Ce(III) taking place on the nanoparticle surface sustains the catalytic and antioxidant activity of the CNPs. This model is based on the chemical structure of the CNP, where the presence of oxygen vacancies on its surface is associated with the presence of Ce(III) sites. This self-regenerating process is of great pharmacological value, as theoretically, CNPs would not exhaust their scavenger activity once inside the cells or other biological systems. This self-sustained mechanism was proposed based on observations made in abiotic systems [52,54]. However, investigations performed in the biological environment evidenced how the oxidation state of the nanoparticle is also greatly influenced by the environment. This aspect has been confirmed by Ce speciation analyses performed on cell-internalized CNPs [34]. Obviously, it cannot be excluded that the CNPs scavenger activity could involve other processes besides their direct oxidation and reduction of free radicals and possibly influence the (integral) proteins’ activity.

The results presented in this study suggest indeed that the overall scavenging action of nanoceria also involves the interaction with AQPs, favoring the H_2_O_2_ membrane transport. After a 2 h incubation, the CNPs are distributed both inside the cells, accumulated in endolysosomes, and on the cell surface, specifically adherent to part of the membrane particularly rich in lipid raft (characterized by cholesterol and sphingolipids) [35]. Sommi et al. demonstrated that the adhesion of CNPs on the membrane is very specific and takes place within minutes even at 4 °C. Such a strong interaction with the component(s) of the membrane could produce an alteration of the membrane characteristics.

The functional experiments showed a double effect of CNPs on osmotic permeability to water: at 15 min incubation, it had an inhibitory effect (reduction by about 50%), while at 2 h it increased by about 60% compared to untreated control cells. The effect was strictly associated with the presence of nanoparticles of cerium oxide. Cells treatment with cerium ions, as Ce(NO_3_)_3_ for both 15 min and 2 h, did not increase the permeability to water but rather decreased it. Since the osmotic permeability to water is considered an index of the permeability of H_2_O_2_ [8,19], it can be assumed that CNPs can somehow regulate the efflux of H_2_O_2_ and the redox cellular status. Soveral et al. using polyoxotungstates demonstrated an inhibitory effect on water permeability similar to that observed by CNPs at 15 min. [55]. However, the effect of CNPs after a 15 min incubation was also studied in Hyper7-NES transfected cells, and the results showed an increased H_2_O_2_ permeability similar to that observed after a 2 h treatment with CNPs (Figure 11). At present, it is difficult to justify the different effects of CNPs on water permeability after 15 min (decrease) and 2 h (increase), especially considering that CNPs increased the H_2_O_2_ permeability in both cases. It can, however, be speculated that at 15 min, CNPs can decrease the permeability by interacting with different AQPs on the outer leaflet of the plasma membrane, some permeable only to water and others to both water and H_2_O_2_. For longer incubations time, when the nanoparticles were already inside the cell, CNPs could activate all AQPs. However, to confirm this hypothesis further experiments are needed in the future.

To prove that the AQP (or AQPs) was the target of the antioxidant effect of CNPs, HeLa cells were selectively silenced for AQP3, AQP6, AQP8, and AQP11. The reduced expression of AQPs was used to measure the transport of H_2_O_2_ in the presence and in the absence of CNPs treatment by means of a Hyper7-NES fluorescent probe. These experiments demonstrated the direct effect of CNPs on AQP-mediated H_2_O_2_ scavenging, confirming, in addition, the results obtained in osmotic water permeability experiments. HeLa cells with a reduced AQP3 expression showed a parallel decrease in H_2_O_2_ permeability but CNPs maintained the ability to stimulate permeability AQP-mediated. AQP8 silencing experiments demonstrated that it was the peroxiporin most involved in H_2_O_2_ membrane transport. The silencing of AQP6 surprisingly caused an increase in the transport of H_2_O_2_, in turn, caused by an upregulation of the AQP8 (see Appendix A). This cellular response would seem to suggest that AQP6 plays an important role in the redox homeostasis of HeLa cells. 

The loss of modification in plasma membrane permeability, observed in AQP11-silenced cells confirmed that this AQP was not involved in the diffusion of H_2_O_2_ across the plasma membrane. This is not surprising given that it has an intracellular localization in the endoplasmic reticulum [10]. However, the possible effect of CNPs on AQP11 will have to be further investigated in the future with an H_2_O_2_ fluorescent probe specifically targeting the ER. 

So far, only one other type of NP has been investigated in relation to AQPs [56,57]. Gold nanoparticles (AuNPs) were shown to modify the AQP1 expression even though the results were at odds with each other, probably due to the different sizes and chemical nature of the nanoparticles and of their surface functionalization. Tiwari et al. found that AuNPs and AuNPs (13 nm size) functionalized with a cell-penetrating peptide downregulated the AQP1 gene in HEp-2 cells [56]. On the contrary, Chen et al. evidenced increased AQP1 levels and endothelial permeability to water that led to brain edema after AuNPs treatment (40 nm size) [57]. Similarly, a completely different agent/effector but comparable in size to NPs, the human papillomavirus, a round small structure with a size ranging from 50 to 60 nm, was capable of reducing the water permeability of human sperm by interacting with AQP8 [21].

As a whole, the results presented here seem to indicate that AQPs are involved in the antioxidant action of cerium nanoparticles. Recently, a number of organic, natural and synthetic, and inorganic antioxidant compounds have been identified as modulators of peroxiporins and oxidative stress [20,24,26]. This suggested the possibility of modulating peroxiporin pores gating, supporting the idea that AQPs may be therapeutic targets. This study directly demonstrates for the first time the effect of nanoparticles on the “gating” of AQPs and, despite the numerous obscure points that will require further experiments, their possible therapeutic use in the treatment of numerous diseases involving oxidative stress.

## 4. Materials and Methods

### 4.1. Cell Culture

HeLa cells were grown in Dulbecco’s modified minimal essential medium–high glucose, supplemented with 10% fetal bovine serum, 1% L-glutamine, 1% penicillin, and streptomycin, and maintained at 37 °C in a humidified atmosphere of 5% CO_2_.

For CNP and cerium nitrate treatments, cells were washed and incubated with PBS containing CNPs (1:10) or with 20 µM final concentration cerium nitrate for 15 min or 2 h.

### 4.2. Cerium Nanoparticles Preparation and Characterization

CNPs were synthetized by direct precipitation from an aqueous solution. They were stabilized by PAA as described previously [34,35]. Briefly, Ce(NO_3_)_3_·6H_2_O (0.1243 g; Merck, Milan, Italy) was dissolved in 50 mL distilled water. PAA (1.22%, Polyscience, Warrington, PA, USA) was added to obtain a 2:1 (*v*/*v*) ratio. A concentrated NH_4_OH solution was added to obtain a pH of about 12 and the solution was kept under constant stirring for 48 h. After centrifugation (13,000× *g*), the supernatant was recovered in deionized H_2_O. The final CNP suspension had a CeO_2_ concentration of ~5 mg/mL, as determined by an inductively coupled plasma–optical emission spectrometry analysis. For the preparation of fluorescent CNPs (Em 580 nm), DiI fluorescent dye was added during the synthesis as previously described [35].

The nanoparticle characterizations were performed as previously described [34,35]. The synthetized CNPs had a hydrodynamic diameter of 14 nm and a Z potential of −20 as evaluated with a Nano ZS90 DLS apparatus (Malvern Instruments, Malvern, UK). The X-ray diffraction (XRD) analysis was performed using a Bruker D8 Advance diffractometer (Bruker Corp., Billerica, MA, USA) operated at 40 kV and 40 mA, showing a dimension of about 5 nm.

### 4.3. RNA Isolation and RT-qPCR

Total RNA was isolated from HeLa cells and reverse-transcribed using, respectively, QIAzol Lysis Reagent (Qiagen, Milan, Italy) and MMLV Reverse Transcriptase M1701 (Promega, Milan, Italy), as previously described [58]. In Table 1 are listed the primers used for AQP1, 2, 3, 4, 5, 6, 7, 8, 9, 10, 11 amplification. Briefly, QuantiFast SYBRGreen PCR Master Mix (Qiagen, Milan, Italy) was used to perform the qPCR. Thermal cycle conditions for qPCR as well as the procedures used were previously reported [36]. Melt curve analysis and separation of the PCR products by gel agarose electrophoresis were performed to verify the presence of single and specific products [36]. Relative mRNA quantitation was expressed as ΔCt, obtained by subtracting the Ct(housekeeping gene) from Ct(AQP gene). Note that high ΔCt values reflect low mRNA expression levels. 

### 4.4. SDS-PAGE and Immunoblotting

Total protein from HeLa cells were treated with a RIPA buffer (150 mM NaCl, 0.5% sodium deoxycholate, 0.1% SDS, 0.1% Triton X-100, 50 mM Tris-HCl, pH 8) supplemented with the protease inhibitor cocktail cOmplete (cOmplete Tablets EASYpack, 04693116001; Merck, Milan, Italy). Total proteins were solubilized in a Laemmli buffer and 30 µg of proteins was separated by SDS-PAGE using precast gel electrophoresis (4–20% Mini-PROTEAN TGX Stain-Free Gels, Bio-Rad, Segrate, Italy) [59] and blotted onto the PVDF Membrane (Trans-Blot Turbo Transfer Pack, #1704156, Bio-Rad, Segrate, Italy) with the Trans-Blot Turbo Transfer System (#1704150, Bio-Rad, Segrate, Italy). After blocking for 1 h at RT in Tris-buffered saline containing 5% nonfat dry milk and 0.1% Tween (blocking solution), membranes were incubated overnight with anti-AQP3 rabbit antibody (PA1488, 1:1000 dilution; BOSTER biological technology, Pleasanton, CA, USA), anti-AQP6 rabbit polyclonal IgG (# AQP61-A, 1:1000 dilution; Alpha Diagnostic International, San Antonio, TX, USA), anti-AQP8 rabbit antibody (HPA046259, 1:500 dilution; Sigma-Aldrich, St. Louis, MO, USA.), and anti-AQP11 rabbit antibody (PA10044, 1:1000 dilution; BOSTER biological technology, Pleasanton, CA, USA), in a blocking solution. As previously described [36], after washing, the membranes were washed and incubated for 1 h with goat antirabbit IgG antibody, peroxidase-conjugated (1:100000; AP132P; Millipore part of Merck S.p.a., Vimodrone, Italy). The bands were detected with Westar Supernova Western blotting detection system (CYANAGEN, Bologna, Italy) and the molecular weights were identified using pre-stained molecular weight markers (ab116028, Abcam, Cambridge, UK). Anti-β-2-microglobulin rabbit monoclonal (ab75853, 1:10000; Abcam, Cambridge, UK) was used on the stripped membrane [60]. Protein bands were detected with the iBright™ CL1000 Imaging System (Thermo Fisher Scientific, Monza, Italy). iBright Analysis Software (Thermo Fisher Scientific, Monza, Italy) was used to perform the semiquantitation analysis of the bands, and the results were expressed as AQP/ β-2-microglobulin ratio.

### 4.5. Double Immunofluorescence

Subconfluent HeLa cells grown on 18 mm × 18 mm glass coverslips were incubated for 2 h at RT with fluorescent CNPs (Em 580 nm) (see Section 4.2). Cells were then fixed in 4% paraformaldehyde in PBS for 30 min in a Petri dish and washed in PBS. After permeabilization by incubating at 80 °C for 30 min with the retrieval buffer (10 mM citrate-HCl, pH 6.0), the samples were blocked with 3% BSA in PBS at RT for 30 min. Coverslips were incubated overnight at 4 °C with the affinity pure anti-AQP3 rabbit antibody (ab125045, 1:400 dilution; Abcam, Cambridge, UK), anti-AQP6 rabbit polyclonal IgG (AQP61-A, 1:200 dilution; Alpha Diagnostic International, San Antonio, TX, USA), anti-AQP8 rabbit antibody (HPA046259, 1:500 dilution; Sigma-Aldrich, St. Louis, MO, U.S.A.), and anti-AQP11 rabbit antibody (ab122821, 1:50 dilution; Abcam, Cambridge, UK). After three 10 min washes with PBS, coverslips were incubated at RT with the fluorescent secondary antibody, Alexa Fluor 647 goat antirabbit (111-606-003, 1:300 dilution; Jackson Immunoresearch Europe Ltd., Cambridgeshire, UK), for 1 h. After three washes with PBS, the coverslips were mounted in ProLong Gold antifade reagent containing 4′,6-Diamino-2-Phenylindole (DAPI; Molecular Probes part of Thermo Fisher Scientific, Monza, Italy), and examined with a TCS SP5 II confocal microscopy system (Leica biosystems, Buccinasco, Italy) equipped with a DM IRBE inverted microscope (Leica biosystems, Buccinasco, Italy). Images acquired with a 60× objective were analyzed by LAS AF Lite software (Leica Microsystems Application Suite Advanced Fluorescence Lite version 2.6.0, Buccinasco, Italy). Cells incubated with nonimmune serum were negative controls.

For colocalization analyses, 3D images were analyzed with Fiji Colocalization Plugin (JACoP) to obtain the Pearson’s correlation coefficient r, Manders’ colocalization coefficient (M1 and M2), and Van Steensel’s cross-correlation function (CCF) [37,38,39].

### 4.6. Gene Silencing

siRNA targeting AQP6 was purchased by Dharmacon™ (ON-TARGETplus Human AQP6 (363) siRNA-SMARTpool, L-011579-00-0005, Dharmacon™, Horizon Discovery Group, Waterbeach, UK). The silencing of AQP3, AQP8, and AQP11 was performed by using predesigned esiRNA from Merck (human AQP3, EHU071641; human AQP8, EHU034911; human AQP11, EHU037771; Merck, Milan, Italy). EHUFLUC esiRNA targeting firefly luciferase was used as a negative control.

AQP3, AQP6, AQP8, and AQP11 were silenced by transfecting the cells with siRNA oligonucleotides or esiRNA, by using the INTERFERin siRNA Transfection Reagent (# 409-10, Polyplus transfection, Illkirch-Graffenstaden, France), as previously described [36]. Briefly, the 50% confluency cell monolayers were treated with fresh medium containing the silencing solution. To prepare the silencing solution, siRNA (25 nM) and siRNA (20 nM) were diluted in Opti-MEM and then mixed with INTERFERin siRNA Transfection Reagent, according to the manufacturer’s instructions. The silencing solution was then diluted in a fresh medium, added to the cells, and incubated at 37 °C for 24 or 48 h.

The effectiveness of the silencing was determined by immunoblotting, and silenced cells were used after 48 h from transfection except for AQP8-silenced cells, which were used after 24 h from transfection.

### 4.7. Water Permeability Measurements

The osmotic water permeability was evaluated by the stopped-flow light scattering method. Experiments were conducted at RT using an RX2000 stopped-flow apparatus (Applied Photophysics, Leatherhead, UK) with a pneumatic trigger accessory (DA.1, Applied Photophysics, Leatherhead, UK) coupled with the Varian Cary 50 spectrophotometer (Varian Australia Pty Ltd., Mulgrave, Australia). The intensity of the scattered light was measured at the wavelength of 450 nm with a dead time of 6 ms. Cell swelling caused by exposure to a hypotonic gradient (150 mOsm/L) was measured for 60 s and an acquisition rate of one reading/0.0125 s. The initial rate constant k was calculated by a computerized least squares regression, fitting the experimental points of the time course of light scattering with a one-phase exponential decay equation (GraphPad Prism 4.00, 2003). 

To study the effect of CNPs on the water permeability, HeLa cells (incubated with and without CNPs) were scraped from the flasks with a scraper, centrifuged, and resuspended in PBS. The osmotic permeability to H2O of AQPs was demonstrated to be indicative of H_2_O_2_ permeability [8]. Five groups were considered: 1. untreated cells (control); 2. cells treated for 15 min with CNPs; 3. cells treated for 15 min with CNPs and subsequently with β-mercaptoethanol (β-ME) 15mM; 4. cells treated for two hours with CNPs; and 5. cells treated for two hours with CNPs and subsequently with the β-ME 15 mM. The same groups were considered to study the effect of cerium nitrate on HeLa cells.

The effect of CNPs on the water permeability of HeLa cells was also examined in oxidative stress conditions. The following six groups were considered: 1. unstressed cells (control); 2. cells incubated for 3 h at 42 °C (endogenous oxidative stress); 3. cells treated for 45 min with 50 μM H_2_O_2_ (exogenous oxidative stress); 4. unstressed cells pretreated for two hours with the CNPs; 5. cells pretreated for two hours with the CNPs, and then incubated for 3 h at 42 °C; and 6. cells pretreated for two hours with the CNPs, and then incubated for 45 min with 50 μM H_2_O_2_.

### 4.8. Hydrogen Peroxide Indicator Transfection for Optical Imaging

The plasmid for the mammalian expression of cytoplasm-targeted ultrasensitive hydrogen peroxide indicator HyPer7 for optical imaging (pCS2+HyPer7-NES) was a generous gift from Vsevolod Belousov (IBCh, Moscow, Russia) (Addgene plasmid # 136467; http://n2t.net/addgene:136467, (accessed on 24 June 2022; RRID: Addgene_136467)) [61]. HyPer7-NES transfection (1 μg DNA/dish) was performed on 60–70% confluent HeLa cells in 2 mL Petri dishes by using the JetOPTIMUS DNA Transfection Reagent (# 117-15, Polyplus transfection, Illkirch-Graffenstaden, France), according to the manufacturer’s instructions. Briefly, Opti-MEM containing the plasmid DNA and the transfection reagent was added to the cells, following the medium removal. A JetOPTIMUS Buffer (# 717-60, Polyplus transfection, Illkirch-Graffenstaden, France) was used to dilute the plasmid DNA (1 μg) and then combined with JetOPTIMUS Reagent to 1:1 ratio (μg of DNA: μL of transfection reagent) and left at RT for 10 min. The solution containing the DNA was added dropwise to the cells and, after 4 h at 37 °C, the medium was removed, and fresh medium added. All the experiments were performed 24 h after transfection. For functional experiments with HeLa silenced cells, Hyper7-NES transfection was performed the following day after the silencing of AQP3, 6, and 11, and the day before the silencing of AQP8 in order to respect the incubation times.

### 4.9. Intracellular H_2_O_2_ Detection by HyPer7-NES Imaging

The effect of CNPs on the H_2_O_2_ permeability Hyper7 oxidation was measured by a ratiometric method [61]. Confocal images were collected every 1–2 s for 1 to 5 min by dual excitation at 420 nm and 490 nm, and the emission was collected at 530 nm. Preliminary experiments showed that results obtained by ratiometric measurements were similar to those obtained by measuring the fluorescence of the HyPer7-NES biosensor excited at 490 nm and the emission collected at 530 nm. For this reason, the following method was routinely used. An Olympus BX41 microscope with a 60× water immersion objective (LUMPlanFI 60×/0.90 w, Olympus) was used to visualize the fluorescence of transfected cells. HyPer7-NES transfected cells were pretreated for two hours with the CNPs, washed with a physiological buffer (140 mM NaCl, 5 mM KCl, 2 mM CaCl_2_, 1 mM MgCl_2_, 10 mM D-glucose, and 1 mM HEPES, pH 7.4) and incubated for 10 min at RT with the same buffer. For live cells fluorescence imaging, the HyPer7-NES biosensor was excited at 490 nm and its emission was collected at 530 nm. Images were acquired using a CCD camera (DMK 33UP1300) and collected at 10 fps by IC capture software. H_2_O_2_ was added to the cells at a final concentration of 50 μM. Image processing was performed with Image J.

### 4.10. Protein Content

The Bradford method was used to determine the protein content [62]. Bovine serum albumin was used as standard.

### 4.11. Statistics

All data were expressed as means ± SEM (Standard Error Mean). Unless otherwise stated in the legends of the figures, the significance of the differences of the means was evaluated by using a one-way ANOVA, followed by Newman–Keuls’s *Q* test, or Student’s *t*-test. All statistical tests were carried out with GraphPad Prism 4.00, 2003.

## Figures and Tables

**Figure 1 ijms-23-10837-f001:**
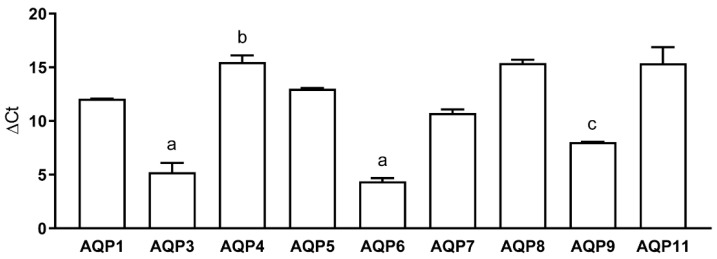
Aquaporins mRNA expression in HeLa cells. Transcripts of all AQPs were found except AQP2 and AQP10. Bars represent the mean  ±  SEM of ΔCt values (n  =  4). ^a^, *p*  <  0.05 versus AQP1, AQP4, AQP5, AQP7, AQP8, AQP11; ^b^, *p*  <  0.05 versus AQP6, AQP9; ^c^, *p* < 0.05 versus AQP8, AQP11.

**Figure 2 ijms-23-10837-f002:**
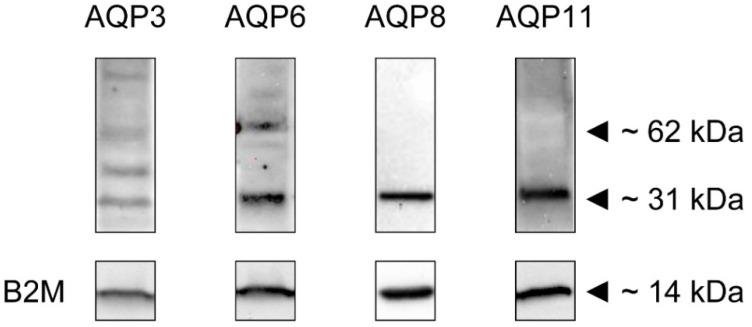
Expression of AQP3, AQP6, AQP8, and AQP11 proteins in HeLa cells. Blots representative of three were shown. Each lane was loaded with 30 μg of proteins and probed with affinity-purified antibodies as described in the Materials and Methods. The same blots were stripped and reprobed with an anti-β-2-microglobulin (β2M) antibody, as housekeeping. Major bands of the expected molecular weights are shown.

**Figure 3 ijms-23-10837-f003:**
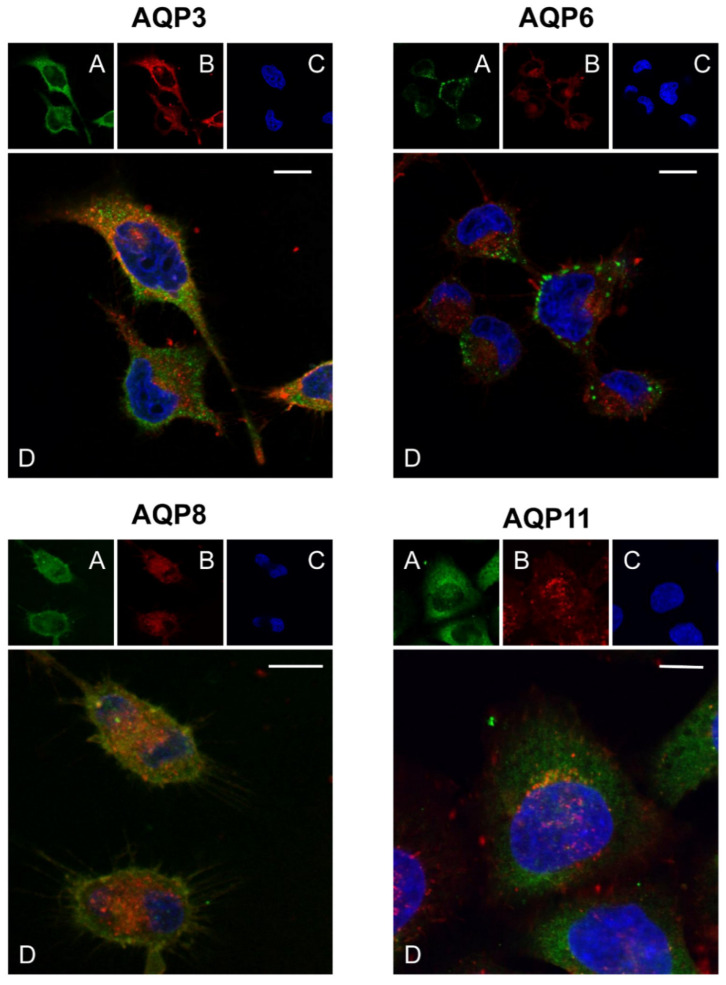
Representative images of confocal laser scanning microscopy of AQP3, AQP6, AQP8, and AQP11 with CNPs in HeLa cells. Green labeling indicates the presence of AQPs (**A**), red labeling the CNPs (**B**), and DAPI (blue; **C**) counterstained nuclei. Yellow labeling shows the colocalization signal of AQP with CNPs (**D**). Scale bar, 10 μm.

**Figure 4 ijms-23-10837-f004:**
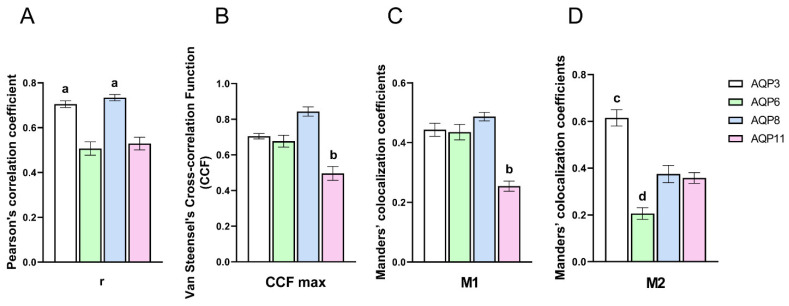
Three-dimensional colocalization analysis of AQP3, AQP6, AQP8, and AQP11 with CNPs in HeLa cells. Statistical analysis of Pearson’s correlation coefficient r (**A**), Van Steensel’s maxima cross-correlation function (CCFmax) (**B**), and Manders’ colocalization coefficients (M1 and M2) (**C**,**D**) were obtained from 4 different double immunofluorescence experiments with anti-AQP antibodies and red-labeled CNPs (see Materials and Methods). Coefficients were determined by 3D analysis of at least 20 cells for each cell line (8–15 z-stack for image) using the JACoP plugin of Fiji. The columns represent the mean ± SEM of the coefficient values (Brown–Forsythe and Welch ANOVA tests followed by Dunnett T3 post-test). ^a^, *p* < 0.05 versus AQP6, AQP11; ^b^, *p* < 0.05 versus AQP3, AQP6, AQP8; ^c^, *p* < 0.05 versus AQP6, AQP8, AQP11; ^d^, *p* < 0.05 versus AQP8, AQP11.

**Figure 5 ijms-23-10837-f005:**
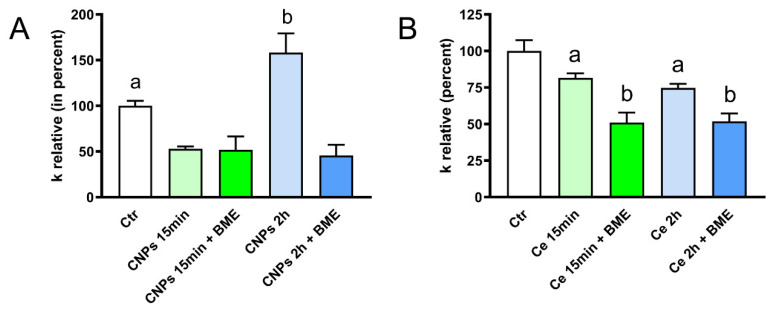
Effect of cerium formulated in nanoparticles (CNPs; **A**) and cerium nonformulated in CNPs (Ce; **Β**) on the water permeability of HeLa cells. Cells were treated for 15 min or 2 h with CNPs and Ce in the presence or in the absence of a post-treatment with β-mercaptoethanol (BME). Cells were then exposed to a 150 mOsm osmotic gradient. Bars represent the osmotic water permeability of HeLa cells expressed as a percent of k relative. Ctr, controls. Values are means ± SEM of 4–15 single shots for each of 4 different experiments (ANOVA, followed by Newman–Keuls’s *Q* test). (**A**) ^a^, *p* < 0.05 versus CNPs 15 min, CNPs 15 min + ΒME, CNPs 2 h, CNPs 2 h + ΒME; ^b^, *p* < 0.05 versus CNPs 15 min, CNPs 15 min + ΒME, CNPs 2 h + ΒME. (**B**) ^a^, *p* < 0.05 versus Ctr, Ce 15 min + ΒME, Ce 2 h + ΒME; ^b^, *p* < 0.05 versus Ctr.

**Figure 6 ijms-23-10837-f006:**
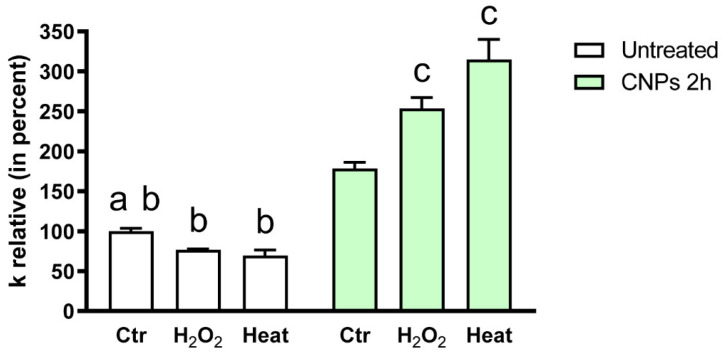
Effect of CNPs on the water permeability of HeLa cells in normal and in oxidative stress conditions. The osmotic water permeability was studied in HeLa cells treated with CNPs (green bars) or in untreated cells (white bars) in three different conditions: (1) unstressed cells (control; Ctr); (2) heat-stressed cells (heat; endogenous oxidative stress); (3) cells treated with H_2_O_2_ (H_2_O_2_; exogenous oxidative stress). Bars represent the osmotic water permeability of HeLa cells expressed as a percent of k relative. Values are means ± SEM of 4–15 single shots for each of 4 different experiments (ANOVA, followed by Newman–Keuls’s *Q* test). ^a^, *p* < 0.05 versus H_2_O_2_, Heat; ^b^, *p* < 0.05 versus CNPs treated cells; ^c^, *p* < 0.05 versus Ctr.

**Figure 7 ijms-23-10837-f007:**
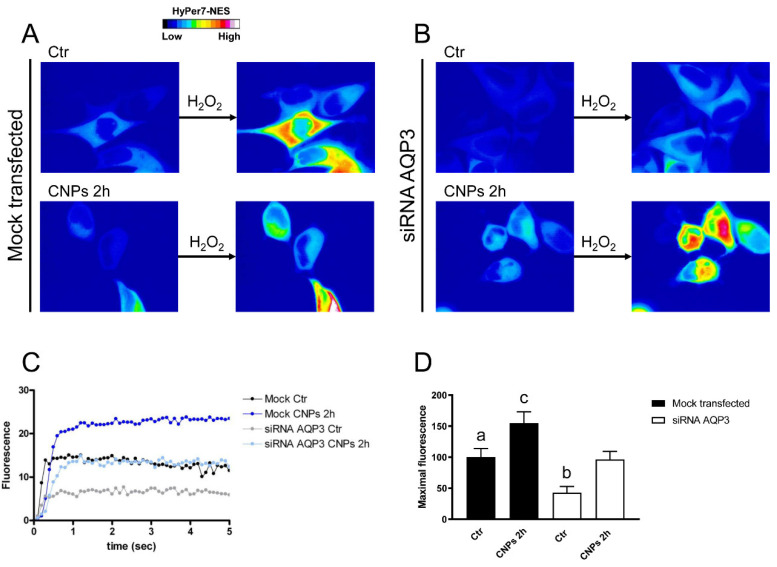
Hydrogen peroxide transport in HeLa cells with AQP3 reduced expression in the presence and in the absence of CNPs treatment. (**A**) Representative frames extracted from videos showing the kinetics of H_2_O_2_ transport into mock-transfected HeLa cells before (left panel) and after (right panel) addition of 50 μM H_2_O_2_ in the absence (Ctr) and in the presence of CNPs treatment (CNPs). The increased HyPer7-NES fluorescence is shown in pseudocolor (upper panel, the scale used is indicated in the insert). (**B**) Representative frames extracted from videos showing the kinetics of H_2_O_2_ transport into AQP3-silenced (siRNA AQP3) HeLa cells before (left panel) and after (right panel) addition of 50 μM H_2_O_2_ in the absence (Ctr) and in the presence of CNPs treatment (CNPs). The increased HyPer7-NES fluorescence is shown in pseudocolor (upper panel, the scale used is indicated in the insert). (**C**) Time course of H_2_O_2_ fluorescence into mock- and siRNA-transfected HeLa cells, in the absence (Ctr) and in the presence of CNPs treatment (CNPs). Curves start when 50 μM H_2_O_2_ was added. Results are the mean of at least 3 different experiments and SEMs were omitted for clarity. (**D**) Maximal H_2_O_2_ fluorescence values were obtained by computerized least squares regression, fitting the experimental points of the time courses of H_2_O_2_ transported curves with a one-phase exponential association equation (GraphPad Prism 4.00, 2003). ^a^, *p* < 0.05 versus CNPs 2 h mock-transfected, control AQP3-null; ^b^, *p* < 0.05 versus CNPs 2 h AQP3-null; ^c^, *p* < 0.05 versus control AQP3-null, CNPs 2 h AQP3-null (ANOVA for repeated measures followed by Newman–Keuls’s *Q* test).

**Figure 8 ijms-23-10837-f008:**
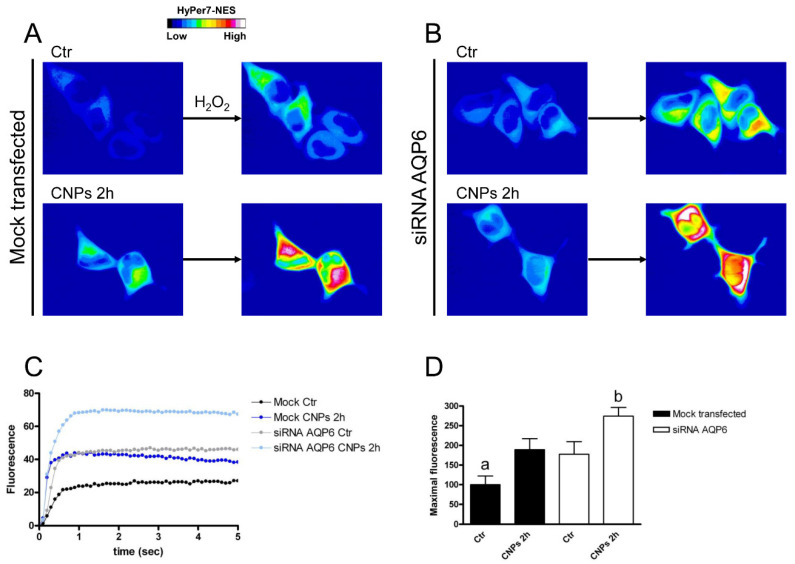
Hydrogen peroxide transport in HeLa cells with AQP6 reduced expression in the presence and in the absence of CNPs treatment. (**A**) Representative frames extracted from videos showing the kinetics of H_2_O_2_ transport into mock-transfected HeLa cells before (left panel) and after (right panel) addition of 50 μM H_2_O_2_ in the absence (Ctr) and in the presence of CNPs treatment (CNPs). The increased HyPer7-NES fluorescence is shown in pseudocolor (upper panel, the scale used is indicated in the insert). (**B**) Representative frames extracted from videos showing the kinetics of H_2_O_2_ transport into AQP6-silenced (siRNA AQP6) HeLa cells before (left panel) and after (right panel) addition of 50 μM H_2_O_2_ in the absence (Ctr) and in the presence of CNPs treatment (CNPs). The increased HyPer7-NES fluorescence is shown in pseudocolor (upper panel, the scale used is indicated in the insert). (**C**) Time course of H_2_O_2_ fluorescence into mock- and siRNA-transfected HeLa cells, in the absence (Ctr) and in the presence of CNPs treatment (CNPs). Curves start when 50 μM H_2_O_2_ was added. Results are the mean of at least 3 different experiments and SEMs were omitted for clarity. (**D**) Maximal H_2_O_2_ fluorescence values were obtained by computerized least squares regression, fitting the experimental points of the time courses of H_2_O_2_ transported curves with a one-phase exponential association equation (GraphPad Prism 4.00, 2003). ^a^, *p* < 0.05 versus control AQP6-null, CNPs 2 h mock-transfected, CNPs 2 h AQP6-null; ^b^, *p* < 0.05 versus control AQP6-null, CNPs 2 h mock-transfected (ANOVA for repeated measures followed by Newman–Keuls’s *Q* test).

**Figure 9 ijms-23-10837-f009:**
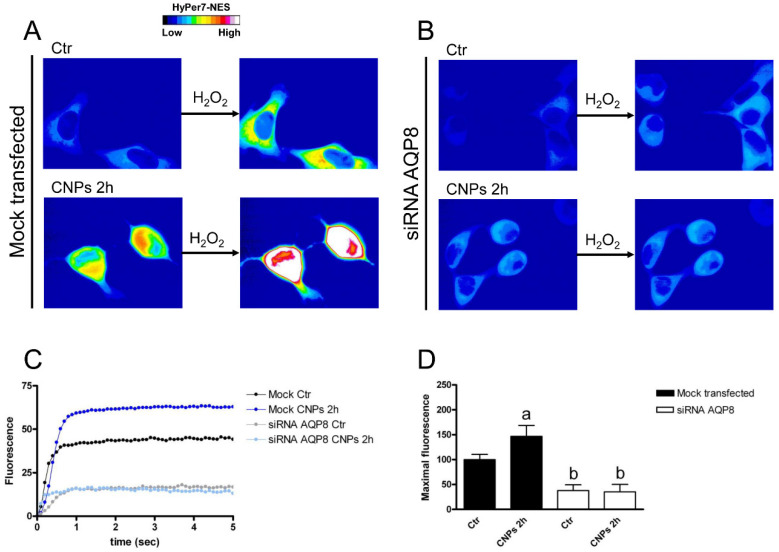
Hydrogen peroxide transport in HeLa cells with AQP8 reduced expression in the presence and in the absence of CNPs treatment. (**A**) Representative frames extracted from videos showing the kinetics of H_2_O_2_ transport into mock-transfected HeLa cells before (left panel) and after (right panel) addition of 50 μM H_2_O_2_ in the absence (Ctr) and in the presence of CNPs treatment (CNPs). The increased HyPer7-NES fluorescence is shown in pseudocolor (upper panel, the scale used is indicated in the insert). (**B**) Representative frames extracted from videos showing the kinetics of H_2_O_2_ transport into AQP8-silenced (siRNA AQP8) HeLa cells before (left panel) and after (right panel) addition of 50 μM H_2_O_2_ in the absence (Ctr) and in the presence of CNPs treatment (CNPs). The increased HyPer7-NES fluorescence is shown in pseudocolor (upper panel, the scale used is indicated in the insert). (**C**) Time course of H_2_O_2_ fluorescence into mock- and siRNA-transfected HeLa cells, in the absence (Ctr) and in the presence of CNPs treatment (CNPs). Curves start when 50 μM H_2_O_2_ was added. Results are the mean of at least 3 different experiments and SEMs were omitted for clarity. (**D**) Maximal H_2_O_2_ fluorescence values were obtained by computerized least squares regression, fitting the experimental points of the time courses of H_2_O_2_ transported curves with a one-phase exponential association equation (GraphPad Prism 4.00, 2003). ^a^, *p* < 0.05 versus Ctr mock-transfected, Ctr AQP8-null, CNPs 2 h AQP8-null; ^b^, *p* < 0.05 versus Ctr mock-transfected (ANOVA for repeated measures followed by Newman–Keuls’s *Q* test).

**Figure 10 ijms-23-10837-f010:**
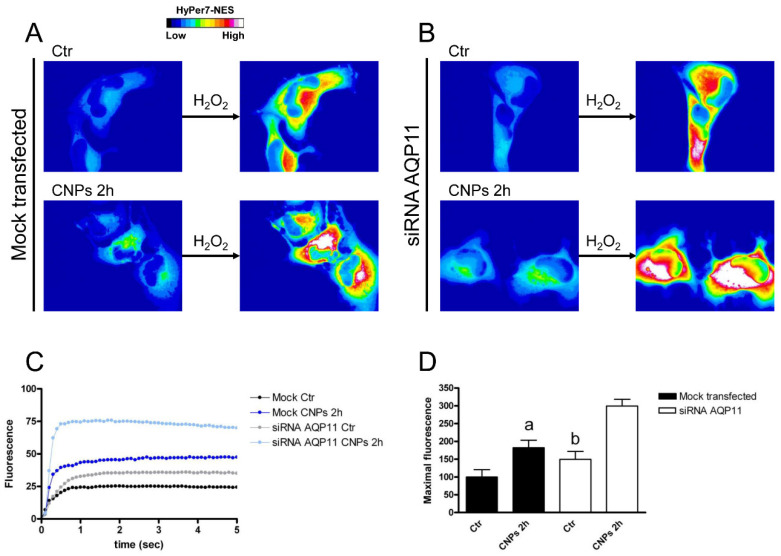
Hydrogen peroxide transport in HeLa cells with AQP11 reduced expression in the presence and in the absence of CNPs treatment. (**A**) Representative frames extracted from videos showing the kinetics of H_2_O_2_ transport into mock-transfected HeLa cells before (left panel) and after (right panel) addition of 50 μM H_2_O_2_ in the absence (Ctr) and in the presence of CNPs treatment (CNPs). The increased HyPer7-NES fluorescence is shown in pseudocolor (upper panel, the scale used is indicated in the insert). (**B**). Representative frames extracted from videos showing the kinetics of H_2_O_2_ transport into AQP11-silenced (siRNA AQP11) HeLa cells before (left panel) and after (right panel) addition of 50 μM H_2_O_2_ in the absence (Ctr) and in the presence of CNPs treatment (CNPs). The increased HyPer7-NES fluorescence is shown in pseudocolor (upper panel, the scale used is indicated in the insert). (**C**) Time course of H_2_O_2_ fluorescence into mock- and siRNA-transfected HeLa cells, in the absence (Ctr) and in the presence of CNPs treatment (CNPs). Curves start when 50 μM H_2_O_2_ was added. Results are the mean of at least 3 different experiments and SEMs were omitted for clarity. (**D**) Maximal H_2_O_2_ fluorescence values were obtained by computerized least squares regression, fitting the experimental points of the time courses of H_2_O_2_ transported curves with a one-phase exponential association equation (GraphPad Prism 4.00, 2003). ^a^, *p* < 0.05 versus Ctr mock-transfected, CNPs 2 h AQP11-null; ^b^, *p* < 0.001 versus CNPs 2 h AQP11-null (ANOVA for repeated measures followed by Newman–Keuls’s *Q* test).

**Figure 11 ijms-23-10837-f011:**
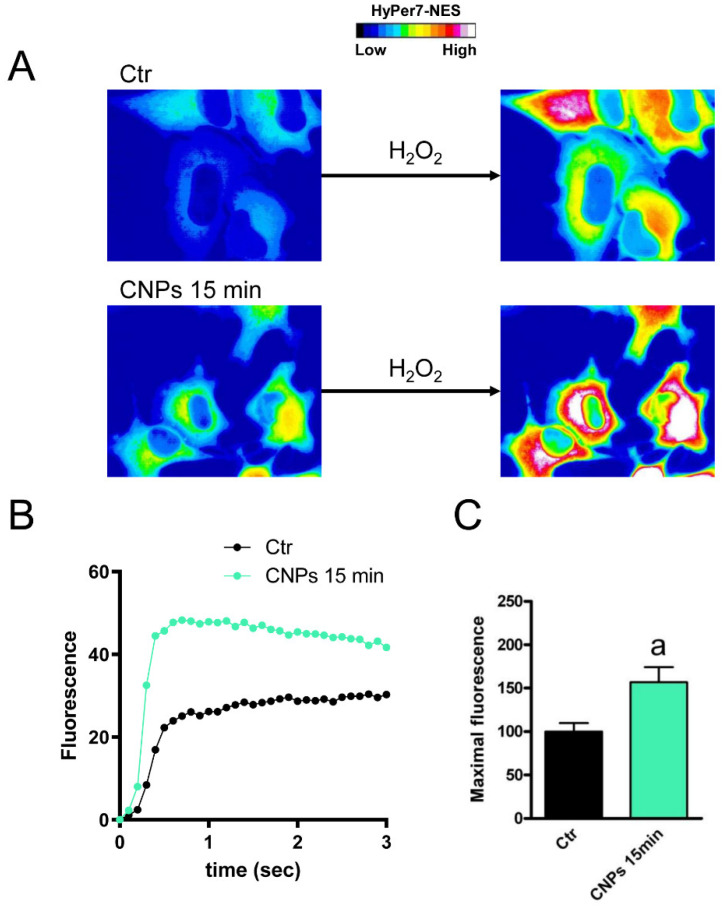
Hydrogen peroxide transport in HeLa cells after a 15 min CNPs treatment. (**A**) Representative frames extracted from videos showing the kinetics of H_2_O_2_ transport into HeLa cells before (left panel) and after (right panel) addition of 50 μM H_2_O_2_ in the absence (Ctr) and in the presence of CNPs treatment (CNPs). The increased HyPer7-NES fluorescence is shown in pseudocolor (upper panel, the scale used is indicated in the insert). (**B**) Time course of H_2_O_2_ fluorescence into HeLa cells, in the absence (Ctr) and in the presence of CNPs treatment (CNPs). Curves start when 50 μM H_2_O_2_ was added. Results are the mean of at least 3 different experiments and SEMs were omitted for clarity. (**C**) Maximal H_2_O_2_ fluorescence values were obtained by computerized least squares regression, fitting the experimental points of the time courses of H_2_O_2_ transported curves with a one-phase exponential association equation (GraphPad Prism 4.00, 2003). ^a^, *p* < 0.05 versus control (Ctr) (Student’s *t*-test).

**Table 1 ijms-23-10837-t001:** Primer sequences are used for real-time reverse-transcription/polymerase chain reaction.

Gene	Primer Sequences		Size (bp)	Accession Number
AQP1 ^a^	Forward	5’-ACACCTCCTGGCTATTGACTACAC-3’	134	NM_198098; variants 1, 5
	Reverse	5’-CCGATGAATGGCCCCACCCAGAA-3’		
AQP2 ^a^	Forward	5’-CACCTCCTTGGGATCCATTACACC-3’	95	NM_000486
	Reverse	5’-ACCCAGTGGTCATCAAATTTGCC-3’		
AQP3 ^b^	Forward	5’-CCTGGTGATGTTTGGCTGTGGCTC-3’	147	NM_004925; variants 1, 2
	Reverse	5’-TTCAGGTGGGCCCCAGAGACC-3’		
AQP4	Forward	5’-GGAGTCACCATGGTTCATGGAA-3’	123	NM_001650; variants 1–3
	Reverse	5’-AGTGACATCAGTCCGTTTGGAA-3’		
AQP5	Forward	5’-GGTGGTGGAGCTGATTCTGA-3’	142	NM_001651
	Reverse	5’-GAAGTAGATTCCGACAAGGTGG-3’		
AQP6	Forward	5’-CACCTCATTGGGATCCACTTC-3’	103	NM_ 001652; variants 1, 2
	Reverse	5’-CCCAGAAGACCCAGTGGACT-3’		
AQP7	Forward	5’-GGACAGCTGATGGTGACCGG-3’	104	NM_001170; variants 1–4
	Reverse	5’-AGCCACGCCTCATTCAGGAA-3’		
AQP8	Forward	5’-TGGAGAGATAGCCATGTGTGAG-3’	106	NM_001169
	Reverse	5’-TGGCTGCACAAACCGTTCGT-3’		
AQP9	Forward	5’-CCCAGCTGTGTCTTTAGCAA-3’	133	NM_020980; variants 1–3
	Reverse	5’-AAGTCCATCATAGTAAATGCCAAA-3’		
AQP10	Forward	5’-CCTATGTTCTCTACCATGATGCCC-3’	137	NM_080429
	Reverse	5’-CTGATCCAGGAAGCCATTGTTC-3’		
AQP11	Forward	5’-TTTCTCTTCCACAGCGCTCT-3’	115	NM_173039; variant 1
	Reverse	5’-CTCCTGTTAGACTTCCTCCTGC-3’		
B2M	Hs_B2M_1_SG QuantiTect Primer Assay QT00088935, Qiagen		98	NM_004048

Melting temperature, 60 °C; ^a^, 62 °C; ^b^, 66 °C.

## Data Availability

Not applicable.

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
