# Peer review of "Cerium Oxide Nanoparticles Regulate Oxidative Stress in HeLa Cells by Increasing the Aquaporin-Mediated Hydrogen Peroxide Permeability"

_ijms, 2022, doi:10.3390/ijms231810837_

Round 1

Reviewer 1 Report

Manuscript prepared by Pellavio G et al in my opinion deserves to be published because it is very interesting and well written paper. It presents the influence of cerium oxide nanoparticles on oxidative stress level in HeLa cell line. Authors studied the influence of CNPs on aquaporin functioning. 

In the Introduction part of the manuscript sufficient information is contained.

The results are presented in a clear, lucid and understandable way. Figures are legible and photos of good quality.

The methodology used is correct and is described adequately and comprehensively.

The discussion presents the confrontation of the results with the literature data in an interesting and comprehensive way.

Conclusions are supported by the data

Author Response

Thank you very much for your comments.

Reviewer 2 Report

Aquaporins have been traditionally associated with the diffusion of water across membranes, but seven aquaporins also facilitate the transport of H2O2. Under certain circumstances, the flux of H2O2 through these proteins is suppressed, ultimately decreasing oxidative stress, thereby acting in an antioxidant role. These particular aquaporins are potential therapeutic targets to control the amount of H2O2 entering cells. This study sought to test whether cerium oxide nanoparticles (CNPs) – which exert general antioxidant effects – control aquaporin efficiency, and subsequently, the redox state of the cell. The results are interesting, and with adjustments, the manuscript could be improved:

1. A rationale for HeLa cells would be helpful – that is, why is this particular cell line best for these studies?

2. In reference to Figure 1, the manuscript states that aquaporins 3, 6, and 9 all have higher expression than the others. Is that indeed accurate?

3. Why does Figure 2 only include four of the aquaporins of interest?

4. In Figure 6: of the cells treated with CNPs, were the H2O2 and heat groups significantly different from the CNP control? Would their letter designation be “a”?

5. Outcomes related to oxidative stress and damage would have been helpful for understanding the broader importance of aquaporins in regulating H2O2.

Author Response

Aquaporins have been traditionally associated with the diffusion of water across membranes, but seven aquaporins also facilitate the transport of H2O2. Under certain circumstances, the flux of H2O2 through these proteins is suppressed, ultimately decreasing oxidative stress, thereby acting in an antioxidant role. These particular aquaporins are potential therapeutic targets to control the amount of H2O2 entering cells. This study sought to test whether cerium oxide nanoparticles (CNPs) – which exert general antioxidant effects – control aquaporin efficiency, and subsequently, the redox state of the cell. The results are interesting, and with adjustments, the manuscript could be improved:

Thank you very much for your comments.

  1. A rationale for HeLa cells would be helpful – that is, why is this particular cell line best for these studies?

We used HeLa cells as they were already used in our previous studies and well characterized in terms of AQPs expression, water and hydrogen peroxide permeability (with or without oxidative stress). The modulation of AQPs pore gating by small compounds was investigated in these cells. Moreover, HeLa cells were also the cells of choice for our Ceria nanoparticles-cell interaction studies.

A sentence has been added in the Introduction to emphasize this point.

Ref.:

  • Sigma-1 Receptor Agonists Acting on Aquaporin-Mediated H2O2 Permeability: New Tools for Counteracting Oxidative Stress. Pellavio G, Rossino G, Gastaldi G, Rossi D, Linciano P, Collina S, Laforenza U. Int J Mol Sci. 2021 Sep 10;22(18):9790.
  • Regulation of Aquaporin Functional Properties Mediated by the Antioxidant Effects of Natural Compounds. Pellavio G, Rui M, Caliogna L, Martino E, Gastaldi G, Collina S, Laforenza U. Int J Mol Sci. 2017 Dec 8;18(12):2665.
  • Stress Regulates Aquaporin-8 Permeability to Impact Cell Growth and Survival. Medraño-Fernandez I, Bestetti S, Bertolotti M, Bienert GP, Bottino C, Laforenza U, Rubartelli A, Sitia R. Antioxid Redox Signal. 2016 Jun 20;24(18):1031-44.
  • Ferraro, D.; Tredici, I.G.; Ghigna, P.; Castillo-Michel, H.; Falqui, A.; Di Benedetto, C.; Alberti, G.; Ricci, V.; Anselmi-Tamburini, U.; Sommi, P. Dependence of the Ce(iii)/Ce(iv) ratio on intracellular localization in ceria nanoparticles internalized by human cells. Nanoscale 2017, 9, 1527-1538.
  • Sommi, P.; Vitali, A.; Coniglio, S.; Callegari, D.; Barbieri, S.; Casu, A.; Falqui, A.; Vigano', L.; Vigani, B.; Ferrari, F.; Anselmi-Tamburini, U. Microvilli Adhesion: An Alternative Route for Nanoparticle Cell Internalization. A.C.S. Nano. 2021, 15, 15803-15814.
  1. In reference to Figure 1, the manuscript states that aquaporins 3, 6, and 9 all have higher expression than the others. Is that indeed accurate?

As indicated in the Materials and methods, mRNA values are expressed as delta Ct, with high delta Ct values reflecting low mRNA expression levels. However, we realize this might not be clear enough to the reader. We have introduced a sentence in the results of the revised paper to clarify this point. 

  1. Why does Figure 2 only include four of the aquaporins of interest?

In Figure 2 are shown only those peroxiporins we found showing bands in western blot and whose protein expression was also confirmed by immunocytochemistry. The explanation was included in the Results.

  1. In Figure 6: of the cells treated with CNPs, were the H2O2 and heat groups significantly different from the CNP control? Would their letter designation be “a”?

We thank the reviewer for pointing out that the lettering indicating the significance was unclear. We modified Figure 6 and the related legend by adding “c” on the H2O2 and heat groups of the cells treated with CNPs to show they are significantly different from the CNP control.

  1. Outcomes related to oxidative stress and damage would have been helpful for understanding the broader importance of aquaporins in regulating H2O2.

The effect of CNPs counteracting the oxidative stress condition by regulating the AQPs gating was demonstrated directly by measuring the water permeability (see Figure 6).

We want to thank the reviewer for the suggestions. In the forthcoming study analyzing in detail the effects of CNPs in compartmentalized AQPs, we will consider the experiments suggested.

Reviewer 3 Report

The article Cerium Oxide Nanoparticles regulate oxidative stress by increasing the aquaporin-mediated hydrogen peroxide permeability by Georgia Pellavio et al. is certainly beneficial for biomedicine. It reveals the potential of CNPs in the treatment of pathological conditions such as neurodegenerative diseases or cancer.

I have the following comments on the manuscript:

1.      As you mentioned in the Introduction, seven AQP paralogs (AQP0, 1, 3, 5, 8, 9, and AQP11) have shown hydrogen peroxide permeability. Why did you focus on AQP 6 in your study then, if according to the title you concentrate on hydrogen peroxide permeability? The decision should be explained in the goals

2.      A, B, C, D symbols are missing in several figures (but it is mentioned in their description).

3.      How do you explain the fact that when monitoring the water permeability of HeLa cells, in the case of cerium formulated in nanoparticles (Fig. 5 A), the permeability decreased after 15 min, but on the contrary, after 2 hours of incubation, the permeability increased significantly compared to the controls? You tried to explain this disproportion in the discussion, but it is still not completely clear to me, so it would be good to consider reformulating of it.

4.      Figure 7 – at first is the description, then the figure. In the entire manuscript it is the other way

In any case, the potential of CNPs is undeniable and of great importance. I recommend the article for acceptance after minor revision

Author Response

Reviewer# 3

The article Cerium Oxide Nanoparticles regulate oxidative stress by increasing the aquaporin-mediated hydrogen peroxide permeability by Georgia Pellavio et al. is certainly beneficial for biomedicine. It reveals the potential of CNPs in the treatment of pathological conditions such as neurodegenerative diseases or cancer.

I have the following comments on the manuscript:

  1. As you mentioned in the Introduction, seven AQP paralogs (AQP0, 1, 3, 5, 8, 9, and AQP11) have shown hydrogen peroxide permeability. Why did you focus on AQP 6 in your study then, if according to the title you concentrate on hydrogen peroxide permeability? The decision should be explained in the goals

As suggested, we introduced a sentence in the aims of the paper explaining the reasons why, in addition to the well-known peroxiporins, we also study AQP6 whose permeability to H2O2 has recently been demonstrated by us (Pellavio, G.; Martinotti, S.; Patrone, M.; Ranzato, E.; Laforenza, U. Aquaporin-6 May Increase the Resistance to Oxidative Stress of Malignant Pleural Mesothelioma Cells. Cells 2022, 11,1892.).

  1. A, B, C, D symbols are missing in several figures (but it is mentioned in their description).

Thank you for pointing it out. In the revised version of the paper, we have introduced the lettering in the figures.

  1. How do you explain the fact that when monitoring the water permeability of HeLa cells, in the case of cerium formulated in nanoparticles (Fig. 5 A), the permeability decreased after 15 min, but on the contrary, after 2 hours of incubation, the permeability increased significantly compared to the controls? You tried to explain this disproportion in the discussion, but it is still not completely clear to me, so it would be good to consider reformulating of it.

The paragraph has been modified trying to describe the concept more clearly but we are aware that at the moment it is pure speculation. To clear this point, we are planning a series of future experiments to assess the effects of CNPs (based on the incubation times), like pulse-chase experiments where the CNP effect would be investigated following  the nanoparticles “synchronized” distribution as opposite to CNPs left available to the cells ad libitum for the entire incubation time.

  1. Figure 7 – at first is the description, then the figure. In the entire manuscript it is the other way

In the revised version of the paper, the description of figure 7 precedes the figure itself.

 In any case, the potential of CNPs is undeniable and of great importance. I recommend the article for acceptance after minor revision.

Thank you very much for your positive comments!

Reviewer 4 Report

The manuscript entitled "Cerium Oxide Nanoparticles regulate oxidative stress by increasing the aquaporin-mediated hydrogen peroxide permeability" brings interesting scientific information regarding the role of AQP in cellular lesions induced by oxidative stress. Therefore I consider the manuscript important for future researches able to open new therapeutic methods  for oxidative stress modulation. 

I suggest to add in the title the HeLa cells, the cells they used for this research in order to be more specific for the readings who are looking for the novelty in this scientific field (the title is to general). The same observation for oxidative stress (be more specific about the oxidative stress parameters you studied - H2O2).

Congratulation for this work! 

Author Response

Reviewer# 4

The manuscript entitled "Cerium Oxide Nanoparticles regulate oxidative stress by increasing the aquaporin-mediated hydrogen peroxide permeability" brings interesting scientific information regarding the role of AQP in cellular lesions induced by oxidative stress. Therefore I consider the manuscript important for future researches able to open new therapeutic methods for oxidative stress modulation. 

I suggest to add in the title the HeLa cells, the cells they used for this research in order to be more specific for the readings who are looking for the novelty in this scientific field (the title is to general). The same observation for oxidative stress (be more specific about the oxidative stress parameters you studied - H2O2).

We modified the title by introducing HeLa cells, as suggested.

Congratulation for this work! 

Thank you very much for your positive comments!

Round 2

Reviewer 2 Report

Prior concerns have been addressed.

Author Response

Many thanks for taking the time to review our paper.